# CONFIDENCE-CONDITIONED VALUE FUNCTIONS FOR OFFLINE REINFORCEMENT LEARNING

## ABSTRACT

Offline reinforcement learning (RL) promises the ability to learn effective policies solely using existing, static datasets, without any costly online interaction. To do so, offline RL methods must handle distributional shift between the dataset and the learned policy. The most common approach is to learn conservative, or lower-bound, value functions, which underestimate the return of out-of-distribution (OOD) actions. However, such methods exhibit one notable drawback: policies optimized on such value functions can only behave according to a fixed, possibly suboptimal, degree of conservatism. However, this can be alleviated if we instead are able to learn policies for varying degrees of conservatism at training time and devise a method to dynamically choose one of them during evaluation. To do so, in this work, we propose learning value functions that additionally condition on the degree of conservatism, which we dub *confidence-conditioned value functions*. We derive a new form of a Bellman backup that simultaneously learns Q-values for any degree of confidence with high probability. By conditioning on confidence, our value functions enable adaptive strategies during online evaluation by controlling for confidence level using the history of observations thus far. This approach can be implemented in practice by conditioning the Q-function from existing conservative algorithms on the confidence. We theoretically show that our learned value functions produce conservative estimates of the true value at any desired confidence. Finally, we empirically show that our algorithm outperforms existing conservative offline RL algorithms on multiple discrete control domains.

## 1 INTRODUCTION

Offline reinforcement learning (RL) aims to learn effective policies entirely from previously collected data, without any online interaction (Levine et al., 2020). This addresses one of the main bottlenecks in the practical adoption of RL in domains such as recommender systems (Afsar et al., 2021), healthcare (Shortreed et al., 2011; Wang et al., 2018), and robotics (Kalashnikov et al., 2018), where exploratory behavior can be costly and dangerous. However, offline RL introduces new challenges, primarily caused by *distribution shift*. Naïve algorithms can grossly overestimate the return of actions that are not taken by the *behavior policy* that collected the dataset (Kumar et al., 2019a). Without online data gathering and feedback, the learned policy will exploit these likely suboptimal actions. One common approach to handle distribution shift in offline RL is to optimize a a conservative lower-bound estimate of the expected return, or Q-values (Kumar et al., 2020; Kostrikov et al., 2021; Yu et al., 2020). By intentionally underestimating the Q-values of out-of-distribution (OOD) actions, policies are discouraged from taking OOD actions. However, such algorithms rely on manually specifying the desired degree of conservatism, which decides how pessimistic the estimated Q-values are. The performance of these algorithms is often sensitive to this choice of hyperparameter, and an imprecise choice can cause such algorithms to fail.

Our work proposes the following solution: instead of learning one pessimistic estimate of Q-values, we propose an offline RL algorithm that estimates Q-values for all possible degrees of conservatism. We do so by conditioning the learned Q-values on its *confidence level*, or probability that it achieves a lower-bound on the true expected returns. This allows us to learn a range of lower-bound Q-values of different confidences. These *confidence-conditioned* Q-values enables us to do something conservative RL algorithms could not—control the level of confidence used to evaluate actions. Specifically, when evaluating the offline-learned Q-values, policies derived from conservative offline

RL algorithms must follow a static behavior, even if the online observations suggest that they are being overly pessimistic or optimistic. However, our approach enables *confidence-adaptive* policies that can correct their behavior using online observations, by simply adjusting the confidence-level used to estimate Q-values. We posit that this adaptation leads to successful policies more frequently than existing static policies that rely on tuning a rather opaque hyperparameter during offline training.

Our primary contribution is a new offline RL algorithm that we call *confidence-conditioned value-learning* (CCVL), which learns a mapping from confidence levels to corresponding lower-bound estimations of the true Q-values. Our theoretical analysis shows that our method learns appropriate lower-bound value estimates for any confidence level. Our algorithm also has a practical implementation that leverages multiple existing ideas in offline RL. Namely, we use network parameterizations studied in distributional RL to predict Q-values parameterized by confidence (Dabney et al., 2018b;a). Our objective, similar to conservative Q-learning (CQL) (Kumar et al., 2020), uses regularization to learn Q-values for all levels of pessimism and optimism, instead of anti-exploration bonuses that may be difficult to accurately compute in complex environments (Rezaeifar et al., 2021). In addition, our algorithm can be easily extended to learn both lower- and upper-bound estimates, which can be useful when fine-tuning our offline-learned value function on additional data obtained via online exploration. Finally, we show that our approach outperforms existing state-of-the-art approaches in discrete-action environments such as Atari (Mnih et al., 2013; Bellemare et al., 2013). Our empirical results also confirm that conditioning on confidence, and controlling the confidence from online observations, can lead to significant improvements in performance.

## 2 RELATED WORK

Offline RL (Lange et al., 2012; Levine et al., 2020) has shown promise in numerous domains. The major challenge in offline RL is distribution shift (Kumar et al., 2019a), where the learned policy might select out-of-distribution actions with unpredictable consequences. Methods to tackle this challenge can be roughly categorized into policy-constraint or conservative methods. Policy-constraint methods regularize the learned policy to be "close" to the behavior policy either explicitly in the objective via a policy regularizer (Fujimoto et al., 2018; Kumar et al., 2019a; Liu et al., 2020; Wu et al., 2019; Fujimoto & Gu, 2021), implicitly update (Siegel et al., 2020; Peng et al., 2019; Nair et al., 2020), or via importance sampling (Liu et al., 2019; Swaminathan & Joachims, 2015; Nachum et al., 2019). On the other hand, conservative methods learn a lower-bound, or conservative, estimate of return and optimize the policy against it (Kumar et al., 2020; Kostrikov et al., 2021; Kidambi et al., 2020; Yu et al., 2020; 2021). Conservative approaches traditionally rely on estimating the epistemic uncertainty, either explicitly via exploration bonuses (Rezaeifar et al., 2021) or implicitly using regularization on the learned Q-values (Kumar et al., 2020). The limitation of existing offline RL approaches is that the derived policies can only act under a fixed degree of conservatism, which is determined by an opaque hyperparameter that scales the estimated epistemic uncertainty, and has to be chosen during offline training. This means the policies will be unable to correct their behavior online, even if it becomes evident from online observations that the estimated value function is too pessimistic or optimistic.

Our algorithm learns confidence-conditioned Q-values that capture all possible degrees of pessimism by conditioning on the confidence level, modeling epistemic uncertainty as a function of confidence. By doing so, instead of committing to one degree of pessimism, we enable policies that adapt how conservative they should behave using the observations they sees during online evaluation. Our approach is related to ensemble (Agarwal et al., 2020; Lee et al., 2021; Chen et al., 2021; An et al., 2021) approaches in that they also predict multiple Q-values to model epistemic uncertainty. However, existing ensemble methods train individual Q-values on the same objective, and rely on different parameter initializations. In contrast, each of our Q-values captures a different confidence-level. In addition, standard ensemble approaches do not consider adaptive policies. Recently, APE-V proposes using ensembles to learn adaptive policies that condition on belief over which value function is most accurate (Ghosh et al., 2022). Our approach considers a similar strategy for adaptation, but explicitly parameterizes the value function by the confidence level, introducing a novel training objective for this purpose. In our experiments, we compare to a method that adapts APE-V to our discrete-action benchmark tasks. Jiang & Huang (2020); Dai et al. (2020) propose confidence intervals for policy evaluation at specified confidence-levels. We aim to learn a value function across all confidences, and use it for adaptive policy optimization. Finally, distributional RL (Dabney et al., 2017; Bellemare et al., 2017; Dabney et al., 2018b) learns a distribution over values, but only capture aleatoric uncertainty, whereas our focus is on epistemic uncertainty and offline RL.

## 3 Preliminaries

The goal in reinforcement learning is to learn a policy $\pi(\cdot|\mathbf{s})$ that maximizes the expected cumulative discounted reward in a Markov decision process (MDP), which is defined by a tuple $(\mathcal{S}, \mathcal{A}, P, R, \gamma)$. $\mathcal{S}, \mathcal{A}$ represent state and action spaces, $P(\mathbf{s}'|\mathbf{s}, \mathbf{a})$ and $R(\mathbf{s}, \mathbf{a})$ represent the dynamics and reward distribution, and $\gamma \in (0, 1)$ represents the discount factor. We assume that the reward $r(\mathbf{s}, \mathbf{a})$ is bounded in magnitude, i.e., $|r(\mathbf{s}, \mathbf{a})| \leq R_{max}$ for some finite $R_{max}$. $\pi_\beta(\mathbf{a}|\mathbf{s})$ represents the (unknown) behavior policy used to collect the offline dataset $\mathcal{D}$ that will be used for training, $d^{\pi_\beta}(\mathbf{s})$ is the discounted marginal state distribution of $\pi_\beta(\mathbf{a}|\mathbf{s})$, and the offline dataset $\mathcal{D} = \{(\mathbf{s}, \mathbf{a}, r, \mathbf{s}')\}$ is formed from interactions sampled from $d^{\pi_\beta}(\mathbf{s})\pi_\beta(\mathbf{a}|\mathbf{s})$.

Policy evaluation attempts to learn the $Q$-function $Q^\pi : \mathcal{S} \times \mathcal{A} \to \mathbb{R}$ of a policy $\pi$ at all state-action pairs $(\mathbf{s}, \mathbf{a}) \in \mathcal{S} \times \mathcal{A}$, Specifically, for a policy $\pi$, its Q-value $Q^\pi(\mathbf{s}, \mathbf{a}) = \mathbb{E}_\pi \left[ \sum_{t=0}^\infty \gamma^t r_t \right]$ is its expected mean return starting from that state and action. The Q-function is the unique fixed point of the Bellman operator $\mathcal{B}^\pi$ given by $\mathcal{B}^\pi Q(\mathbf{s}, \mathbf{a}) = r(\mathbf{s}, \mathbf{a}) + \gamma \mathbb{E}_{\mathbf{s}' \sim P(\mathbf{s}'|\mathbf{s}, \mathbf{a}), \mathbf{a}' \sim \pi(\mathbf{a}'|\mathbf{s}')} [Q(\mathbf{s}', \mathbf{a}')]$, meaning $Q^\pi = \mathcal{B}^\pi Q^\pi$. Q-learning learns $Q^* = Q^{\pi^*}$ as the fixed point of the Bellman optimality operator $\mathcal{B}^*$ given by $\mathcal{B}^* Q(\mathbf{s}, \mathbf{a}) = r(\mathbf{s}, \mathbf{a}) + \gamma \mathbb{E}_{\mathbf{s}' \sim P(\mathbf{s}'|\mathbf{s}, \mathbf{a})} [\max_{\mathbf{a}'} Q(\mathbf{s}', \mathbf{a}')]$, and derives the optimal policy $\pi^*(\mathbf{a} \mid \mathbf{s}) = \mathbb{I}\{\mathbf{a} = \arg\max_{\mathbf{a}} Q^*(\mathbf{s}, \mathbf{a})\}$.

**Offline reinforcement learning.** In offline RL, we are limited to interactions that appear in the dataset $\mathcal{D}$ of $N$ samples $(\mathbf{s}, \mathbf{a}, r, \mathbf{s}')$, where $\mathbf{a} \in \mathcal{A}$ is derived from some suboptimal behavior policy. Hence, we do not have access to the optimal actions used in the backup of the Bellman optimality operator. Because of this, offline RL suffers from distributional shift (Kumar et al., 2019b; Levine et al., 2020). Prior methods address this issue by learning conservative, or lower-bound, value functions that underestimate expected return outside of the dataset. One method to accomplish this is to subtract *anti-exploration bonuses* that are larger for out-of-distribution (OOD) states and actions (Rezaeifar et al., 2021):

$$\widehat{Q}^{k+1} = \arg\min_Q \frac{1}{2} \mathbb{E}_{\mathbf{s}, \mathbf{a}, \mathbf{s}' \sim \mathcal{D}} \left[ \left( Q(\mathbf{s}, \mathbf{a}) - \widehat{\mathcal{B}}^* \widehat{Q}^k(\mathbf{s}, \mathbf{a}) - \alpha \sqrt{\frac{1}{n(s, a) \wedge 1}} \right)^2 \right], \qquad (1)$$

where $\alpha > 0$ is a hyperparameter. Another relevant method is conservative Q-learning (CQL) (Kumar et al., 2020), which proposes a regularizer to the standard objective to learn pessimistic Q-values:

$$\widehat{Q}^{k+1} = \arg\min_Q \max_\pi \ \alpha \left( \mathbb{E}_{\mathbf{s} \sim \mathcal{D}, \mathbf{a} \sim \pi(\mathbf{a}|\mathbf{s})} [Q(\mathbf{s}, \mathbf{a})] - \mathbb{E}_{\mathbf{s}, \mathbf{a} \sim \mathcal{D}} [Q(\mathbf{s}, \mathbf{a})] \right) \qquad (2)$$

$$+ \frac{1}{2} \mathbb{E}_{\mathbf{s}, \mathbf{a}, \mathbf{s}' \sim \mathcal{D}} \left[ \left( Q(\mathbf{s}, \mathbf{a}) - \widehat{\mathcal{B}}^* \widehat{Q}^k(\mathbf{s}, \mathbf{a}) \right)^2 \right] + \mathcal{R}(\pi).$$

Here, $\pi$ is some policy that approximately maximizes the current Q-function iterate, and $\mathcal{R}$ is some regularizer. This objective includes a penalty that ensures Q-values at OOD actions are underestimated compared to in-distribution (ID) actions. Such methods learn lower-bound value functions for a fixed confidence-level, that is implicitly captured in hyperparameter $\alpha$. In this paper, we propose learning value functions that condition the confidence-level explicitly.

**Additional notation.** Let $n \wedge 1 = \max\{n, 1\}$. Denote $\iota = \text{polylog}(|\mathcal{S}|, (1 - \gamma)^{-1}, N)$. We let $\iota$ be a polylogarithmic quantity, changing with context.

## 4 Confidence-Conditioned Value Functions

In this section, we describe our method for learning *confidence-conditioned value functions*, such that conditioned on some confidence level $\delta \in (0, 1)$, the learned Q-function can lower-bound its true value with probability $1 - \delta$. Because such Q-functions depend not only on state-action pairs, but also the confidence $\delta$, they enable adaptive policies that change behavior based on $\delta$, and adjust delta to maximize online performance. In contrast, pessimistic offline RL is limited to a fixed Markovian strategy. We first propose a novel Q-learning algorithm, which we dub *confidence-conditioned value learning* (CCVL), then show how such learned Q-function enables adaptive strategies, dubbed *confidence-adaptive policies*. In this work, we focus on discrete-action environments, but our insights can be straightforwardly extended to develop actor-critic algorithms for continuous environments.

### 4.1 Confidence-Conditioned Value Learning

Recall from Section 3 that standard Q-learning involves learning Q-values that satisfy the Bellman optimality update $Q^* = \mathcal{B}^* Q^*$. We are interested in learning confidence-conditioned Q-values, which we define as:

**Definition 4.1.** *A confidence-conditioned value function $Q(\mathbf{s}, \mathbf{a}, \delta)$ satisfies, for a given $\delta \in (0, 1)$:*

$$Q(\mathbf{s}, \mathbf{a}, \delta) = \sup \ q \quad \textit{such that} \quad \Pr[Q^*(\mathbf{s}, \mathbf{a}) \geq q] \geq 1 - \delta\,. \tag{3}$$

Note that we include the suprenum to prevent $Q(\mathbf{s}, \mathbf{a}, \delta) = Q(\mathbf{s}, \mathbf{a}, 0)$ for all other values of $\delta$. To achieve a high-probability lower-bound on $Q^*(\mathbf{s}, \mathbf{a})$, we account for two sources of uncertainty: (1) we must approximate the Bellman optimality operator, which assumes known reward and transition model, using samples in $\mathcal{D}$, and (2) we need to additionally lower-bound the target $Q^*$ used in the Bellman backup. The uncertainty due to (1), also called *epistemic uncertainty*, can be bounded using concentration arguments on the samples from $\mathcal{D}$. Namely, we define $b(\mathbf{s}, \mathbf{a}, \delta)$ as a high-probability *anti-exploration bonus* that upper-bounds epistemic uncertainty, or $\mathbb{P}\left(\left|\mathcal{B}^* Q^*(\mathbf{s}, \mathbf{a}) - \widehat{\mathcal{B}}^* Q^*(\mathbf{s}, \mathbf{a})\right| \leq b(\mathbf{s}, \mathbf{a}, \delta)\right) \geq 1 - \delta$. Such bonuses are well-studied in the prior literature (Burda et al., 2018; Rezaeifar et al., 2021), and can be derived using concentration inequalities such as Chernoff-Hoeffding or Bernstein. Using the former, the bonuses are given by $b(\mathbf{s}, \mathbf{a}, \delta) = \sqrt{\frac{\iota \log(1/\delta)}{n(\mathbf{s},\mathbf{a}) \wedge 1}}$, where $n(\mathbf{s}, \mathbf{a})$ is the number of times the state-action pair appears in $\mathcal{D}$. Next, the uncertainty due to (2) can be straightforwardly bounded using our learned Q-function. This gives rise to the iterative update for training the confidence-conditioned Q-function:

$$\widehat{Q}^{k+1} = \arg\min_Q \frac{1}{2} \mathbb{E}_{\mathbf{s},\mathbf{a},\mathbf{s}' \sim D}\left[\left(Q(\mathbf{s}, \mathbf{a}, \delta) - \max_{\delta_1, \delta_2 \leq \delta} \widehat{\mathcal{B}}^* \widehat{Q}^k(\mathbf{s}, \mathbf{a}, \delta_2) - \alpha\sqrt{\frac{\log(1/\delta_1)}{n(\mathbf{s}, \mathbf{a}) \wedge 1}}\right)^2\right]\,, \tag{4}$$

where $\alpha > 0$ is again some hyperparameter. In Theorem 6.1, we show that for any confidence level $\delta \in (0, 1)$, the resulting Q-values $\widehat{Q}(\mathbf{s}, \mathbf{a}, \delta) = \lim_{k \to \infty} \widehat{Q}^k(\mathbf{s}, \mathbf{a}, \delta)$ lower-bounds the true Q-value $Q^*(\mathbf{s}, \mathbf{a})$ with probability at least $1 - \delta$.

Note that Equation 4 is similar to a traditional Q-learning using anti-exploration bonuses, as in Equation 1, but with important differences. In conservative Q-learning, the $\delta$ value is not modeled and implicitly captured in the $\alpha$ hyperparameter. Equation 1 can be made more similar to Equation 4 by explicitly conditioning on $\delta$, and setting $\delta_1 = \delta_2 = \delta$. We believe our approach offers the following advantages compared to using anti-exploration bonuses without conditioning. First, tuning $\alpha$ in our approach is easier as we do not need to commit to a degree of conservatism beforehand. Also, by introducing an outer maximization over $\delta_1, \delta_2$, we see that for any iteration $k \in \mathbb{N}$, and any $\delta \in (0, 1)$, $\widehat{Q}^{k+1}(\mathbf{s}, \mathbf{a}, \delta)$ as the solution to Equation 4 is at least as tight of a lower-bound one that would set $\delta_1 = \delta_2 = \delta$. The latter is what Equation 1 implicitly does.

**Implicit bonuses via regularization.** The objective in Equation 4 requires explicit computation of anti-exploration bonuses, which requires computation of state-action visitations $n(\mathbf{s}, \mathbf{a})^{-1}$ that we discuss in Section 5 is difficult with neural network value functions. Here, we propose a new objective that is inspired by how CQL achieves pessimistic value functions (Kumar et al., 2020). The key idea is, instead of explicitly subtracting a bonus, we can add a regularizer in the objective. Specifically, we have the following iterative update as an alternative to equation 4:

$$\widehat{Q}^{k+1} = \arg\min_Q \max_{\delta_1, \delta_2 \leq \delta} \max_\pi \alpha\sqrt{\frac{\log(1/\delta_1)}{(n(\mathbf{s}) \wedge 1)}} \left(\mathbb{E}_{\mathbf{s} \sim D, \mathbf{a} \sim \pi(\mathbf{a}|s)}\left[Q(\mathbf{s}, \mathbf{a}, \delta)\right] - \mathbb{E}_{\mathbf{s}, \mathbf{a} \sim D}\left[Q(\mathbf{s}, \mathbf{a}, \delta)\right]\right)$$
$$+ \frac{1}{2}\mathbb{E}_{\mathbf{s},\mathbf{a},\mathbf{s}' \sim D}\left[\left(Q(\mathbf{s}, \mathbf{a}, \delta) - \widehat{\mathcal{B}}^* \widehat{Q}^k(\mathbf{s}, \mathbf{a}, \delta_2)\right)^2\right] + \mathcal{R}(\pi)\,, \tag{5}$$

where like in Kumar et al. (2020), $\mathcal{R}$ is some regularizer (typically the entropy of $\pi$). Note that Equation 5 still relies on the computation of $n(\mathbf{s})^{-1}$. However, we noticed that estimating state visitations is actually much easier than state-action visitations with neural networks: we observed that state-action density estimators were insufficiently discriminative between seen and unseen actions at a given state, although state-only visitations, which do not require estimating densities

of unseen samples were a bit more reliable (see Section 5 for details) In Theorem 6.2, we show that the resulting $\widehat{Q}(\mathbf{s}, \mathbf{a}, \delta)$ may not point-wise lower-bound $Q^*(\mathbf{s}, \mathbf{a})$, but will do so in expectation. Specifically, for $\widehat{V}(\mathbf{s}, \delta) = \max_{\mathbf{a}} \widehat{Q}(\mathbf{s}, \mathbf{a}, \delta)$, we have that $\widehat{V}(\mathbf{s}, \delta)$ lower-bounds the true value $V^*(\mathbf{s}) = \max_{\mathbf{a}} Q^*(\mathbf{s}, \mathbf{a})$ with probability at least $1 - \delta$.

The objective in Equation 5 differs from the CQL update in Equation 2 in two notable aspects: (1) we explicitly condition on $\delta$ and introduce a maximization over $\delta_1, \delta_2$, and (2) rather than a fixed weight of $\alpha > 0$ on the CQL regularizer, the weight now depends on the state visitations. Like with Equation 4, we can argue that (1) implies that for any $k \in \mathbb{N}$, we learn at least as tight lower-bounds for any $\delta$ than the CQL update implicitly would. In addition, (2) means that the lower-bounds due to the CQL regularizer additionally depends on state visitations in $\mathcal{D}$, which will improve the quality of the obtained lower-bounds over standard CQL.

### 4.2 CONFIDENCE-ADAPTIVE POLICIES

Given a learned Q-function, standard Q-learning would choose a stationary Markovian policy that selects actions according to $\widehat{\pi}(\mathbf{a} \mid \mathbf{s}) = \mathbb{I}\left\{\mathbf{a} = \arg\max_{\mathbf{a}} \widehat{Q}(\mathbf{s}, \mathbf{a})\right\}$. We can naïvely do this with the learned confidence-conditioned Q-function by fixing $\delta$ and tuning it as a hyper-parameter. However, especially in offline RL, it can be preferable for the agent to change its behavior upon receiving new observations during online evaluation, as such observations can show that the agent has been behaving overly pessimistic or optimistic. This adaptive behavior is enabled using our confidence-conditioned Q-function by adjusting $\delta$ using online observations.

Let $h$ be the history of observations during online evaluation thus far. We propose a *confidence-adaptive policy* that conditions the confidence $\delta$ under which it acts on $h$; namely, we propose a non-Markovian policy that selects actions as $\widehat{\pi}(\mathbf{a} \mid \mathbf{s}, h) = \mathbb{I}\left\{\mathbf{a} = \arg\max_{\mathbf{a}} \widehat{Q}(\mathbf{s}, \mathbf{a}, \delta)\right\}$, where $\delta \sim \mathbf{b}(h)$. Here, $\mathbf{b}(h)$ is a distribution representing the "belief" over which $\delta$ is best to evaluate actions for history $h$. Inspired by Ghosh et al. (2022), we compute $\mathbf{b}(h)$ using Bellman consistency (Xie et al., 2021) as a surrogate log-likelihood. Here, the probability of sampling $\delta$ under $\mathbf{b}(h)$ is:

$$\mathbf{b}(h)(\delta) \propto \sum_{(\mathbf{s}, \mathbf{a}, r, \mathbf{s}') \in h} \left(\widehat{Q}(\mathbf{s}, \mathbf{a}, \delta) - r - \gamma \max_{\mathbf{a}'} \widehat{Q}(\mathbf{s}', \mathbf{a}', \delta)\right)^2 \tag{6}$$

Note that this surrogate objective is easy to update with new observations. This leads to a tractable confidence-adaptive policy $\widehat{\pi}$ that can outperform existing Markovian policies that can only act according to a fixed level of pessimism.

### 4.3 LEARNING LOWER- AND UPPER-BOUNDS

A natural extension of our method is to learn confidence-conditioned upper-bounds on the true Q-values. Formally, as change of notation, let $Q_\ell(\mathbf{s}, \mathbf{a}, \delta)$ be the lower-bounds as defined in Equation 3. We can learn upper-bounds $Q_u(\mathbf{s}, \mathbf{a}, \delta)$ as

$$Q_u(\mathbf{s}, \mathbf{a}, \delta) = \inf q \quad \text{s.t} \quad \Pr[Q^*(\mathbf{s}, \mathbf{a}) \leq q] \geq 1 - \delta. \tag{7}$$

Following analogous logic as in Secton 4.1, we can derive an iterative update as

$$\widehat{Q}_u^{k+1} = \arg\min_Q \frac{1}{2} \mathbb{E}_{\mathbf{s}, \mathbf{a}, \mathbf{s}' \sim D}\left[\left(Q(\mathbf{s}, \mathbf{a}, \delta) - \min_{\delta_1, \delta_2 \leq \delta} \widehat{\mathcal{B}}^* \widehat{Q}_u^k(\mathbf{s}, \mathbf{a}, \delta_2) + \alpha \sqrt{\frac{\log(1/\delta_1)}{n(\mathbf{s}, \mathbf{a}) \wedge 1}}\right)^2\right]. \tag{8}$$

Learning both $\widehat{Q}_\ell$ and $\widehat{Q}_u$ presents the opportunity for improved policy extraction from the learned value functions. Instead of simply optimizing the learned lower-bounds, which may lead to overly conservative behavior, we can optimize the upper-bounds but constrained to *safe* actions whose corresponding lower-bounds are not too low. Formally, our policy can perform

$$\widehat{\pi}(\mathbf{a} \mid \mathbf{s}, h) = \mathbb{I}\left\{\mathbf{a} = \arg\max_{\mathbf{a} \in \mathcal{A}_\ell} \widehat{Q}_u(\mathbf{s}, \mathbf{a}, \delta)\right\},$$

$$\text{where } \delta \sim \mathbf{b}(h), \text{ and } \mathcal{A}_\ell = \left\{\mathbf{a} : \widehat{Q}_\ell(\mathbf{s}, \mathbf{a}, \delta) \geq \beta \max_{\mathbf{a}'} \widehat{Q}_\ell(\mathbf{s}, \mathbf{a}', \delta)\right\}, \tag{9}$$

for some parameter $\beta > 0$. To simplify notation, for the remainder of the paper, we again drop the subscript on $\ell$ when referencing lower-bounds. Learning upper-bounds offline is particularly important when fine-tuning the value functions on online interactions, which is a natural next-step after performing offline RL. Existing offline RL algorithms achieve strong offline performance, but lack the exploration necessary to improve greatly during online fine-tuning. By learning both lower- and upper-bounds, our method can achieve better online policy improvement (see Section 7.3).

## 5    PRACTICAL ALGORITHM

In this section, we describe implementation details for our CCVL algorithm, and arrive at a practical algorithm. We aim to resolve the following details: (1) how the confidence-conditioned Q-function is parameterized, and (2) how the objective in Equation 4 or Equation 5 is estimated and optimized.

Our Q-function is parameterized by a neural network with parameters $\theta$. To handle conditioning on $\delta$, we build upon implicit quantile networks (IQN) (Dabney et al., 2018a), and propose a parametric model that can produce $\widehat{Q}(\mathbf{s}, \mathbf{a}, \delta)$ for given values of $\delta$. Alternatively, we could fix quantized values of $\delta$, and model our Q-function as an ensemble where each ensemble member corresponds to one fixed $\delta$. We choose the IQN parameterization because training over many different $\delta \sim \mathcal{U}(0, 1)$ may lead to better generalization over confidences. However, when computing beliefs $\mathbf{b}$ online, we maintain a categorical distribution over quantized values of $\delta$.

In Equation 4 or Equation 5, we must compute the inverse state-action or state visitations. This can be exactly computed for tabular environments. However, in non-tabular ones, we need to estimate inverse counts $n(\mathbf{s}, \mathbf{a})^{-1}$ or $n(\mathbf{s})^{-1}$. In prior work, O'Donoghue et al. (2018) proposed obtaining linear-value estimates using the last layer of the neural network, *i.e.*, $n(\mathbf{s}, \mathbf{a})^{-1} \approx \phi(\mathbf{s})^{\top} \left( \Phi_a^{\top} \Phi_a \right)^{-1} \phi(\mathbf{s})$, where $\phi$ extracts state representations, and $\Phi_a$ is a matrix of $\phi(\mathbf{s}_i)$ for states $\mathbf{s}_i \in \mathcal{D}$ where action $\mathbf{a}$ was taken. However, we found that such methods were not discriminative enough to separate different actions in the dataset from others under the same state. Instead of state-action visitations, the update in Equation 5 requires only estimating inverse state visitations $n(\mathbf{s})^{-1}$. Empirically, we find that linear estimates such as $n(\mathbf{s})^{-1} \approx \phi(\mathbf{s})^{\top} \phi(\mathbf{s})$ could successfully discriminate between states. Hence, we use the latter update when implementing CCVL in non-tabular environments.

Finally, we summarize our CCVL algorithm in Algorithm 1. Note that aside from sampling multiple $\delta \sim \mathcal{U}(0, 1)$ for training, CCVL is no more computationally expensive than standard Q-learning, and is on the same order as distributional or ensemble RL algorithms that train on multiple Q-value estimations per state-action pair. Hence, our algorithm is very practical, while enabling adaptive non-Markovian policies as described in Section 4.2.

---

**Algorithm 1** Confidence-Conditioned Value Learning (CCVL)

**Require:** Offline dataset $\mathcal{D}$, discount factor $\gamma$, weight $\alpha$, number of samples $N, M$
 1: Initialize Q-function $\widehat{Q}_\theta$
 2: **for** step $t = 1, 2, \ldots, n$ **do**
 3:     **for** $i = 1, 2, \ldots, N$ **do**
 4:         Sample confidence $\delta \sim \mathcal{U}(0, 1)$
 5:         For $j = 1, \ldots, M$, sample $\delta_{j,1}, \delta_{j,2} \sim \mathcal{U}(0, \delta)$. Compute $\mathcal{L}_j(\theta)$ as inner-term of right-hand side of equation 4 or equation 5 with $\delta_1 = \delta_{j,1}, \delta_2 = \delta_{j,2}$
 6:         Take gradient step $\theta_t := \theta_{t-1} - \eta \nabla_\theta \max_j \mathcal{L}_j(\theta)$
 7: Return Q-function $\widehat{Q}_\theta$

---

## 6    THEORETICAL ANALYSIS

In this section, we show that in a tabular MDP, the value functions learned by CCVL properly estimate lower-bounds of the true value, for any confidence $\delta$. We show this for both the update using anti-exploration bonuses in equation 4 as well as the one using regularization in Equation 5.

First, we show a simple lemma that CCVL will learn a value function such that the values decrease as the confidence level increases. Formally, we show the following:

**Lemma 6.1.** *The Q-values $\widehat{Q}$ learned via CCVL satisfy, for any $\delta, \delta' \in (0,1)$ such that $\delta \leq \delta'$: $\widehat{Q}(\mathbf{s}, \mathbf{a}, \delta) \leq \widehat{Q}(\mathbf{s}, \mathbf{a}, \delta')$.*

*Proof.* Let $\delta_1, \delta_2 \leq \delta$ be the solution to the maximization for $\widehat{Q}(\mathbf{s}, \mathbf{a}, \delta)$ in Equation 4. Since $\delta \leq \delta'$, we have $\delta_1, \delta_2 \leq \delta'$. This implies $\widehat{Q}(\mathbf{s}, \mathbf{a}, \delta) \leq \widehat{Q}(\mathbf{s}, \mathbf{a}, \delta')$, as desired. $\square$

Lemma 6.1 means that as $\delta$ decreases, which equates to estimating a lower bound of higher confidence, our estimated Q-values will monotonically decrease. Using Lemma 6.1 allows us to show the following theorems, which are the main results of this section. We state the results below, and defer proofs to the Appendix A.

The first shows that using when equation 4, our value-function estimates, for any confidence $\delta \in (0,1)$, a proper lower-bound on the optimal Q-values with probability at least $1 - \delta$.

**Theorem 6.1.** *For any $\delta \in (0,1)$, the Q-values $\widehat{Q}$ learned via CCVL with Equation 4 satisfies:*

$$\widehat{Q}(\mathbf{s}, \mathbf{a}, \delta) \leq Q^*(\mathbf{s}, \mathbf{a}) \quad \textit{for all states } \mathbf{s} \in \mathcal{S} \textit{ and actions } \mathbf{a} \in \mathcal{A},$$

*with probability at least $1 - \delta$ for some $\alpha > 0$.*

The second theorem shows an analogous result to Theorem 6.1, but using the update in equation 5 instead. However, using the alternative update does not guarantee a pointwise lower-bound on Q-values for all state-action pairs. However, akin to Kumar et al. (2020), we can show a lower-bound on the values for all states.

**Theorem 6.2.** *For any $\delta \in (0,1)$, the value of the policy $\widehat{V}(\mathbf{s}, \delta) = \max_{\mathbf{a} \in \mathcal{A}} \widehat{Q}(\mathbf{s}, \mathbf{a}, \delta)$, where $\widehat{Q}$ are learned via CCVL with Equation 5 satisfies:*

$$\widehat{V}(\mathbf{s}, \delta) \leq V^*(\mathbf{s}) \quad \textit{for all states } \mathbf{s} \in \mathcal{S},$$

*where $V^*(\mathbf{s}) = \max_{\mathbf{a} \in \mathcal{A}} Q^*(\mathbf{s}, \mathbf{a})$ with probability at least $1 - \delta$ for some $\alpha > 0$.*

## 7 EMPIRICAL EVALUATION

In our experiments, we aim to evaluate our algorithm, CCVL on discrete-action offline RL tasks. We aim to ascertain whether the two distinct properties of our method lead to improved performance: (1) conditioning on confidence $\delta$ during offline training, and (2) adapting the confidence value $\delta$ during online rollouts. We compare to prior offline RL methods, REM (Agarwal et al., 2020) and CQL (Kumar et al., 2020), and ablations of our method where we either replace confidence-conditioning with a simple ensemble, which we dub *adaptive ensemble value-learning* (AEVL), or behave according to a fixed confidence online, which we call Fixed-CCVL.

**Comparisons.** REM and CQL are existing state-of-the-art offline RL algorithms for discrete-action environments. AEVL allows us to study question (1) by replacing confidence-conditioned values with a random ensemble, where each model in the ensemble roughly has the same level of conservatism. Each ensemble member of AEVL is trained independently using different initial parameters, and which ensemble member to act under is controlled online using Bellman error as in our proposed method. Note that AEVL can be viewed as a special case of APE-V in Ghosh et al. (2022) for discrete-action domains. Finally, Fixed-CCVL tests (2) by treating confidence $\delta$ used by the policy as a fixed hyper-parameter instead of automatically adjusting it during online rollouts. The confidence is selected as the one that minimized Bellman error during offline training. Because AEVL and CCVL change their behavior during evaluation, we maintain a fair comparison by reporting the average score across the adaptation process, including episodes where adaptation has not yet converged.

### 7.1 ILLUSTRATIVE EXAMPLE ON GRIDWORLD

We first present a didactic example that illustrates the benefit of CCVL over standard conservative offline RL algorithms. We consider a $8 \times 8$ gridworld environment (Fu et al., 2019), with a start and goal state, walls, lava. The reward is 1 upon reaching the goal, but entering a lava state results in receiving a reward of 0 for the rest of the trajectory. The gridworld is visualized in Figure 1.

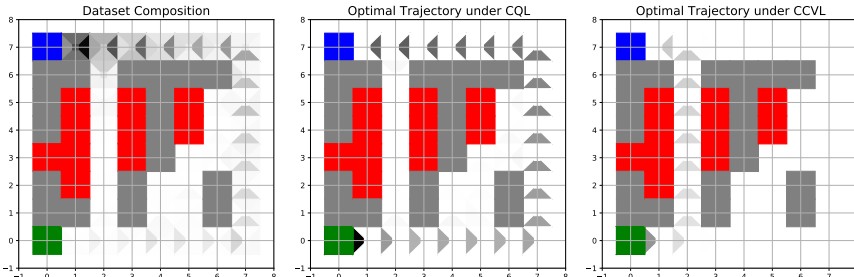

Figure 1: Example gridworld where CQL takes the longer, suboptimal trajectory that appears more frequently in the dataset, but CCVL ultimately adapts $\delta$ and takes the optimal one.

We consider an offline RL task where the learned policy must generalize to a slightly different gridworld environment than the one it was trained on. In our case, during offline training, the environment is stochastic, in that there is a 30% chance that the agent travels in an unintended direction; however, during evaluation, that probability decreases to 15%. This makes previously risky paths more optimal. This is where we anticipate that adaptive methods such as ours will have a severe advantage. While CQL will act too conservatively, our method CCVL can evaluate and change its level of conservatism on the fly.

We construct an offline dataset consisting of 2.5k samples from a behavior policy, which takes the optimal action with probability 0.5, and a random action otherwise. As shown in Figure 1(left), the most visited trajectory in the offline dataset is the safe, longer one that avoids the lava. However, in evaluation, the optimal trajectory is the short one that narrowly misses the lava. In Figure 1, we see that CQL is unable to recover the optimal trajectory whereas our method CCVL can after 10 trajectories of adaptation. In the figure on the right, we visualize the process under which $\delta$ adapts, ultimately becoming less conservative.

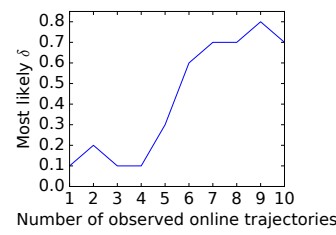

## 7.2 OFFLINE TRAINING ON ATARI

Next, we evaluate our algorithm against prior methods on Atari games (Bellemare et al., 2013) with offline datasets of varying size and quality, previously considered by Agarwal et al. (2020); Kumar et al. (2020). We follow the exact setup of Kumar et al. (2022), including evaluating across the same set of 17 games, using the same three offline datasets, with 1% and 5% of samples uniformly drawn from DQN replay dataset introduced in Agarwal et al. (2020), as well as a more suboptimal dataset consisting of 10% of the initial samples from the DQN dataset (corresponding to the first 20M observations during online DQN). Including this more suboptimal dataset allows us to evaluate the degree to which each method can improve over the average performance in the dataset. Following Agarwal et al. (2020), the Atari games have stochastic dynamics, with a 25% chance of "sticky actions," *i.e.*, executing the previous action instead of a new one.

The REM and CQL baselines use exactly the hyperparamter configurations used by Kumar et al. (2022). We refer to Table E.1 of Kumar et al. (2022) for a table of hyperparamters used. For CCVL, we use the iterative update in Equation 5 as it achieves significantly better performance. Across all methods, we found it useful to perform DR3 regularization on the learned state representations (Kumar et al., 2022). Following Agarwal et al. (2021), we report the interquartile mean (IQM) normalized scores, where the normalization gives score 0 to a random policy and 100 to the nature DQN (Mnih et al., 2015), and each score is computed using the average of 100 episodes. We also report 95% confidence intervals (CIs) computed using stratified bootstrapping. The results across all 17 games for the three datasets are in Table 1. We also show complete per-game results in Tables 4-6.

Note that our method CCVL outperforms all baselines that we evaluate against. Though the average improvement across all games is small, we see that CCVL sometimes outperforms REM and CQL by over 30% for games such as Asterix or Breakout. We believe this is because REM and CQL can only act according to a fixed level of conservatism across all games, whereas CCVL is able to adapt its level on a per-game basis. We also notice that CCVL outperforms both ablations, showing that both confidence-conditioning and adaptation are important to the success of our algorithm. Though

| Data | REM | CQL | AEVL | Fixed-CCVL | CCVL |
|---|---|---|---|---|---|
| 1% | 16.5 | 56.9 | 15.2 | 56.2 | **59.1** |
| | (14.5, 18.6) | (52.5, 61.2) | (53.0, 60.8) | (52.0, 61.4) | **(51.8, 65.6)** |
| 5% | 60.2 | 105.7 | 57.2 | 105.9 | **110.1** |
| | (101.9, 110.9) | (55.8, 65.1) | (50.9, 63.6) | (102.3, 109.9) | **(101.2, 117.4)** |
| Initial 10% | 73.8 | 65.8 | 75.3 | 64.7 | **77.8** |
| | (69.3, 78) | (63.3, 68.3) | (68, 79.5) | (62.7, 67.9) | **(69.1, 87.2)** |

Table 1: Final performance across 17 Atari games after 6.25M gradient updates on 1% data and 12.5M for 5%, 10% in terms of normalized IQM across 5 random seeds, with 95% stratified bootstrap CIs in parentheses. REM and CQL results are from Kumar et al. (2022). Our method CCVL outperforms prior baselines and ablations across all three datasets.

AEVL is adaptive, because the ensemble members do not represent diverse hypotheses about how to act optimally, adaptation is not useful. Perhaps unsurprisingly, Fixed-CCVL and CQL perform similarly due to the similarities in the objective in Equation 5 and Equation 2. However, CCVL greatly improves over Fixed-CCVL due to being able to adapt the $\delta$ used by the policy online.

### 7.3 ONLINE FINE-TUNING ON ATARI

It is often realistic to consider that the value functions obtained by offline RL can be improved additional online interactions, which we call online fine-tuning. Our CCVL method, when extended to learn both lower- and upper-bounds as discussed in Section 4.3, is well-suited for this setting. This is because our approach can leverage lower-bounds to act pessimistically offline, while using upper-bounds for online exploration. Note that unlike the experiments in the previous section, where the CVVL policies adapt $\delta$ during each online rollout, these experiments include additional training with online RL for *all* methods. Again, all methods receive the same exact amount of data, but must now perform online exploration themselves during the fine-tuning phase.

We select 5 representative Atari games, similarly considered in Kumar et al. (2020). We first run offline training across all algorithms on the 1% dataset for 6.25M gradient steps, then run 625k steps of online RL, and report the final performance. We report the gain in normalized IQM after online fine-tuning in Table 2. Our method, CCVL, achieves the best score across 4 of 5 games. Though CQL often achieves the second best overall score, it often sees the smallest improvement, likely due to its conservatism being detrimental to online exploration.

| Game | REM | CQL | CCVL |
|---|---|---|---|
| Asterix | $4.3 \rightarrow 45.2$ | $18.6 \rightarrow 52.7$ | $\mathbf{25.9 \rightarrow 159.5}$ |
| Breakout | $\mathbf{1.2 \rightarrow 204.2}$ | $2.8 \rightarrow 193.7$ | $2.7 \rightarrow 202.7$ |
| Pong | $36.4 \rightarrow 113.4$ | $100.0 \rightarrow 111.6$ | $\mathbf{105.2 \rightarrow 117.9}$ |
| Seaquest | $13.9 \rightarrow 51.2$ | $24 \rightarrow 60.7$ | $\mathbf{30.9 \rightarrow 77.8}$ |
| Qbert | $3.4 \rightarrow 120.4$ | $111.2 \rightarrow 118.9$ | $\mathbf{111.3 \rightarrow 139.7}$ |

Table 2: Improvement in normalized IQM final performance after 625k additional gradient steps of online fine-tuning on top of offline training on 1% data.

### 8 CONCLUSION

In this work, we propose confidence-conditioned value learning (CCVL), a offline RL algorithm that learns a value function for all degrees of conservatism, called confidence-levels. Contrary to standard offline RL algorithms like CQL that must specify a degree of conservatism during training via hyperparameter tuning, CCVL enables condition-adaptive policies that adjust this degree using online observations. CCVL can be implemented practically, using slight modifications on top of existing offline RL algorithms. Theoretically, we show that in a tabular environment, CCVL, for any confidence-level, learns appropriate value that is a lower-bound at that confidence. Empirically, we demonstrate that in discrete-action environments, CCVL perform better than prior methods. We view CCVL as a first-step in proposing conservative offline RL algorithms that adjust their level of conservatism, rather than having the level tuned beforehand via an opaque hyperparameter. Many angles for further investigation exist. Theoretically, it remains to see whether the confidence-conditioned values are lower-bounds under function approximation. Algorithmically, an important direction of future work is to extend CCVL to continuous-action environments, which would involve developing an actor-critic algorithm using confidence-conditioned policies.

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

# A PROOFS

In this section, we provide proofs of theorems stated in Section 6. Recall from Section 3 that $\iota = \text{polylog}(|\mathcal{S}|, (1-\gamma)^{-1}, N)$ is some constant. Our proofs rely on the following lemma, which bounds the estimation error due to using the empirical Bellman operator:

**Lemma A.1.** *For all state-action $(\mathbf{s}, \mathbf{a}) \in \mathcal{S} \times \mathcal{A}$ such that $n(\mathbf{s}, \mathbf{a}) \geq 1$, function Q, and $\delta \in (0,1)$, we have:*

$$\mathbb{P}\left( \left| \widehat{\mathcal{B}}^* Q(\mathbf{s}, \mathbf{a}) - \mathcal{B}^* Q(\mathbf{s}, \mathbf{a}) \right| \leq \sqrt{\frac{\iota \log(1/\delta)}{n(\mathbf{s}, \mathbf{a})}} \right) \geq 1 - \delta \,.$$

The above lemma is a well-known result in reinforcement learning (Rashidinejad et al., 2021), whose derivation follows from Hoeffding's inequalities.

## A.1 PROOF OF THEOREM 6.1

Without loss of generality, assume that $\delta_1, \delta_2 \leq \delta$ are the solution to the outer maximization of Equation 4 at convergence. Using Lemma A.1, we have that

$$\widehat{Q}(\mathbf{s}, \mathbf{a}, \delta) = \widehat{\mathcal{B}}^* \widehat{Q}(\mathbf{s}, \mathbf{a}, \delta_2) - \alpha \sqrt{\frac{\log(1/\delta_1)}{n(\mathbf{s}, \mathbf{a}) \wedge 1}}$$

$$\leq \mathcal{B}^* \widehat{Q}(\mathbf{s}, \mathbf{a}, \delta_2) - \alpha \sqrt{\frac{\log(1/\delta_1)}{n(\mathbf{s}, \mathbf{a}) \wedge 1}} + \sqrt{\frac{\iota \log(1/\delta_1)}{n(\mathbf{s}, \mathbf{a})}} \leq \mathcal{B}^* \widehat{Q}(\mathbf{s}, \mathbf{a}, \delta_2) \quad \forall \mathbf{s} \in \mathcal{S}, \mathbf{a} \in \mathcal{A} \,,$$

holds with probability at least $1 - \delta_1$ for any $\alpha \geq \iota^{1/2}$. Using Lemma 6.1, we have

$$\widehat{Q}(\mathbf{s}, \mathbf{a}, \delta) \leq \mathcal{B}^* \widehat{Q}(\mathbf{s}, \mathbf{a}, \delta) \Longrightarrow \widehat{Q} \leq (I - \gamma P^*)^{-1} R$$
$$\Longrightarrow \widehat{Q}(\mathbf{s}, \mathbf{a}) \leq Q^*(\mathbf{s}, \mathbf{a}) \quad \forall \mathbf{s} \in \mathcal{S}, \mathbf{a} \in \mathcal{A} \,,$$

holds with probability at least $1 - \delta_1 \geq 1 - \delta$, as desired.

## A.2 PROOF OF THEOREM 6.2

Recall from Equation 5 that at convergence, we have,

$$\widehat{Q}(\mathbf{s}, \mathbf{a}, \delta) = \arg\min_Q \max_{\delta_1, \delta_2} \max_\pi \alpha \sqrt{\frac{\log(1/\delta_1)}{(n(\mathbf{s}) \wedge 1)}} \left( \mathbb{E}_{\mathbf{s} \sim D, \mathbf{a} \sim \pi(\mathbf{a}|s)} \left[ Q(\mathbf{s}, \mathbf{a}, \delta) \right] - \mathbb{E}_{\mathbf{s}, \mathbf{a} \sim D} \left[ Q(\mathbf{s}, \mathbf{a}, \delta) \right] \right)$$

$$+ \frac{1}{2} \mathbb{E}_{\mathbf{s}, \mathbf{a}, \mathbf{s}' \sim D} \left[ \left( Q(\mathbf{s}, \mathbf{a}, \delta) - \widehat{\mathcal{B}}^* \widehat{Q}(\mathbf{s}, \mathbf{a}, \delta_2) \right)^2 \right] + \mathcal{R}(\pi)$$

$$\leq \max_{\delta_1, \delta_2} \max_\pi \arg\min_Q \alpha \sqrt{\frac{\log(1/\delta_1)}{(n(\mathbf{s}) \wedge 1)}} \left( \mathbb{E}_{\mathbf{s} \sim D, \mathbf{a} \sim \pi(\mathbf{a}|s)} \left[ Q(\mathbf{s}, \mathbf{a}, \delta) \right] - \mathbb{E}_{\mathbf{s}, \mathbf{a} \sim D} \left[ Q(\mathbf{s}, \mathbf{a}, \delta) \right] \right)$$

$$+ \frac{1}{2} \mathbb{E}_{\mathbf{s}, \mathbf{a}, \mathbf{s}' \sim D} \left[ \left( Q(\mathbf{s}, \mathbf{a}, \delta) - \widehat{\mathcal{B}}^* \widehat{Q}(\mathbf{s}, \mathbf{a}, \delta_2) \right)^2 \right] + \mathcal{R}(\pi)$$

For any $\delta_1, \delta_2 \leq \delta$ and $\pi$, we have that the solution to the inner-minimization over $Q$ yields

$$\tilde{Q}(\mathbf{s}, \mathbf{a}, \delta, \delta_1, \delta_2, \pi) = \arg\min_Q \alpha \sqrt{\frac{\log(1/\delta_1)}{(n(\mathbf{s}) \wedge 1)}} \left( \mathbb{E}_{\mathbf{s} \sim D, \mathbf{a} \sim \pi(\mathbf{a}|s)} \left[ Q(\mathbf{s}, \mathbf{a}, \delta) \right] - \mathbb{E}_{\mathbf{s}, \mathbf{a} \sim D} \left[ Q(\mathbf{s}, \mathbf{a}, \delta) \right] \right)$$

$$+ \frac{1}{2} \mathbb{E}_{\mathbf{s}, \mathbf{a}, \mathbf{s}' \sim D} \left[ \left( Q(\mathbf{s}, \mathbf{a}, \delta) - \widehat{\mathcal{B}}^* \widehat{Q}(\mathbf{s}, \mathbf{a}, \delta_2) \right)^2 \right]$$

$$\leq \widehat{\mathcal{B}}^* \widehat{Q}(\mathbf{s}, \mathbf{a}, \delta_2) - \alpha \sqrt{\frac{\log(1/\delta_1)}{n(\mathbf{s})}} \left[ \frac{\pi(\mathbf{a} \mid \mathbf{s})}{\pi_\beta(\mathbf{a} \mid \mathbf{s})} - 1 \right] \,.$$

This arises from taking the derivative of the minimization objective, and solving for $Q$ that makes the derivative equal to 0. Note that we can simplify

$$\alpha\sqrt{\frac{\log(1/\delta_1)}{n(\mathbf{s})}}\left[\frac{\pi(\mathbf{a}\mid\mathbf{s})}{\pi_\beta(\mathbf{a}\mid\mathbf{s})}-1\right]=\alpha\sqrt{\frac{\log(1/\delta_1)}{n(\mathbf{s})}}\left[\frac{\pi(\mathbf{a}\mid\mathbf{s})-\pi_\beta(\mathbf{a}\mid\mathbf{s})}{\pi_\beta(\mathbf{a}\mid\mathbf{s})}\right]$$

$$=\alpha\sqrt{\frac{\log(1/\delta_1)}{n(\mathbf{s},\mathbf{a})}}\left[\frac{\pi(\mathbf{a}\mid\mathbf{s})-\pi_\beta(\mathbf{a}\mid\mathbf{s})}{\sqrt{\pi_\beta(\mathbf{a}\mid\mathbf{s})}}\right].$$

Without loss of generality, assume that $\delta_1,\delta_2\leq\delta$ and $\pi$ are the solution to the outer-maximization. Substituting the previous result into the equation for $\widehat{Q}(\mathbf{s},\mathbf{a},\delta)$, and applying Lemma A.1 yields,

$$\widehat{Q}(\mathbf{s},\mathbf{a},\delta)\leq\widehat{\mathcal{B}}^*\widehat{Q}(\mathbf{s},\mathbf{a},\delta_2)-\alpha\sqrt{\frac{\log(1/\delta_1)}{n(\mathbf{s},\mathbf{a})}}\left[\frac{\pi(\mathbf{a}\mid\mathbf{s})-\pi_\beta(\mathbf{a}\mid\mathbf{s})}{\sqrt{\pi_\beta(\mathbf{a}\mid\mathbf{s})}}\right]$$

$$\leq\widehat{\mathcal{B}}^*\widehat{Q}(\mathbf{s},\mathbf{a},\delta_2)-\alpha\sqrt{\frac{\log(1/\delta_1)}{n(\mathbf{s},\mathbf{a})}}\left[\frac{\pi(\mathbf{a}\mid\mathbf{s})-\pi_\beta(\mathbf{a}\mid\mathbf{s})}{\sqrt{\pi_\beta(\mathbf{a}\mid\mathbf{s})}}\right]+\sqrt{\frac{\iota\log(1/\delta_1)}{n(\mathbf{s},\mathbf{a})}}.$$

Note that the middle term is not positive if $\pi(\mathbf{a}\mid\mathbf{s})<\pi_\beta(\mathbf{a}\mid\mathbf{s})$. However, we know that for $\mathbf{a}^*=\arg\max_{\mathbf{a}}\widehat{Q}(\mathbf{s},\mathbf{a},\delta)$ then $\pi(\mathbf{a}\mid\mathbf{s})\geq\pi_\beta(\mathbf{a}\mid\mathbf{s})$ by definition of $\pi$ maximizing the learned Q-values. Therefore, we have

$$\widehat{V}(\mathbf{s},\delta)=\widehat{Q}(\mathbf{s},\mathbf{a}^*,\delta)\leq\widehat{\mathcal{B}}^*\widehat{Q}(\mathbf{s},\mathbf{a}^*,\delta_2)-\alpha\sqrt{\frac{\log(1/\delta_1)}{n(\mathbf{s},\mathbf{a}^*)}}\left[\frac{\pi(\mathbf{a}^*\mid\mathbf{s})-\pi_\beta(\mathbf{a}^*\mid\mathbf{s})}{\sqrt{\pi_\beta(\mathbf{a}^*\mid\mathbf{s})}}\right]+\sqrt{\frac{\iota\log(1/\delta_1)}{n(\mathbf{s},\mathbf{a}^*)}}$$

$$\leq\widehat{\mathcal{B}}^*\widehat{V}(\mathbf{s},\delta_2)\quad\forall\mathbf{s}\in\mathcal{S}$$

holds with probability at least $1-\delta_1$ for $\alpha$ satisfying

$$\alpha\geq\iota^{1/2}\max_{\mathbf{s},\mathbf{a}}\left[\frac{\pi(\mathbf{a}\mid\mathbf{s})-\pi_\beta(\mathbf{a}\mid\mathbf{s})}{\sqrt{\pi_\beta(\mathbf{a}\mid\mathbf{s})}}\right]^{-1}.$$

Then, using Lemma A.1, we have

$$\widehat{V}(\mathbf{s},\delta)\leq\widehat{\mathcal{B}}^*\widehat{V}(\mathbf{s},\delta)\implies\widehat{V}(\mathbf{s},\delta)\leq V^*(\mathbf{s})\quad\forall\mathbf{s}\in\mathcal{S},$$

holds with probability at least $1-\delta_1\geq1-\delta$, as desired.

# B  ATARI RESULTS

In this section, we provide per-game results across all Atari games that we evaluated on for the three considered dataset sizes. As mentioned in the main paper, we use the hyperparameter configuration detailed in Kumar et al. (2022) for our Atari experiments. For completion, we also reproduce the table in this section.

Table 3: Hyperparameters used by the offline RL Atari agents in our experiments. We follow the setup of Agarwal et al. (2020); Kumar et al. (2022).

| Hyperparameter | Setting (for both variations) |
|---|---|
| Sticky actions | Yes |
| Sticky action probability | 0.25 |
| Grey-scaling | True |
| Observation down-sampling | (84, 84) |
| Frames stacked | 4 |
| Frame skip (Action repetitions) | 4 |
| Reward clipping | [-1, 1] |
| Terminal condition | Game Over |
| Max frames per episode | 108K |
| Discount factor | 0.99 |
| Mini-batch size | 32 |
| Target network update period | every 2000 updates |
| Training environment steps per iteration | 250K |
| Update period every | 4 environment steps |
| Evaluation $\epsilon$ | 0.001 |
| Evaluation steps per iteration | 125K |
| $Q$-network: channels | 32, 64, 64 |
| $Q$-network: filter size | $8 \times 8, 4 \times 4, 3 \times 3$ |
| $Q$-network: stride | 4, 2, 1 |
| $Q$-network: hidden units | 512 |

| Game | REM | CQL | AEVL | Fixed-CCVL | CCVL |
|---|---|---|---|---|---|
| Asterix | $405.7 \pm 46.5$ | $821.4 \pm 75.1$ | $421.2 \pm 67.8$ | $874.0 \pm 64.3$ | $1032.1 \pm 86.7$ |
| Breakout | $14.3 \pm 2.8$ | $32.0 \pm 3.2$ | $7.4 \pm 1.9$ | $28.7 \pm 2.8$ | $31.2 \pm 4.3$ |
| Pong | $-7.7 \pm 6.3$ | $14.2 \pm 3.3$ | $-8.4 \pm 6.8$ | $14.7 \pm 3.8$ | $15.8 \pm 4.4$ |
| Seaquest | $293.3 \pm 191.5$ | $446.6 \pm 26.9$ | $320.6 \pm 154.1$ | $422.0 \pm 21.9$ | $551.2 \pm 42.2$ |
| Qbert | $436.3 \pm 111.5$ | $9162.7 \pm 993.6$ | $294.6 \pm 100.3$ | $9172.3 \pm 907.6$ | $9170.1 \pm 1023.5$ |
| SpaceInvaders | $206.6 \pm 77.6$ | $351.9 \pm 77.1$ | $224.2 \pm 84.7$ | $355.7 \pm 80.2$ | $355.4 \pm 81.1$ |
| Zaxxon | $2596.4 \pm 1726.4$ | $1757.4 \pm 879.4$ | $2467.8 \pm 2023.4$ | $1747.6 \pm 894.3$ | $2273.6 \pm 1803.1$ |
| YarsRevenge | $5480.2 \pm 962.3$ | $16011.3 \pm 1409.0$ | $4857.1 \pm 1012.6$ | $15890.7 \pm 1218.2$ | $20140.5 \pm 2022.8$ |
| RoadRunner | $3872.9 \pm 1616.4$ | $24928.7 \pm 7484.5$ | $5048.3 \pm 2156.5$ | $22590.3 \pm 6860.2$ | $26780.5 \pm 10112.3$ |
| MsPacman | $1275.1 \pm 345.6$ | $2245.7 \pm 193.8$ | $1164.7 \pm 508.2$ | $2542.3 \pm 188.4$ | $2673.2 \pm 226.4$ |
| BeamRider | $522.9 \pm 42.2$ | $617.9 \pm 25.1$ | $600.1 \pm 57.3$ | $645.3 \pm 40.1$ | $630.2 \pm 37.8$ |
| Jamesbond | $157.6 \pm 65.0$ | $460.5 \pm 102.0$ | $114.3 \pm 56.7$ | $462.1 \pm 98.4$ | $452.1 \pm 153.9$ |
| Enduro | $132.4 \pm 16.1$ | $253.5 \pm 14.2$ | $103.2 \pm 10.1$ | $244.8 \pm 20.9$ | $274.5 \pm 23.8$ |
| WizardOfWor | $1663.7 \pm 417.8$ | $904.6 \pm 343.7$ | $1640.7 \pm 383.4$ | $1488.1 \pm 450.9$ | $1513.8 \pm 652.1$ |
| IceHockey | $-9.1 \pm 5.1$ | $-7.8 \pm 0.9$ | $-10.4 \pm 4.9$ | $-7.6 \pm 1.1$ | $-7.1 \pm 1.6$ |
| DoubleDunk | $-17.6 \pm 1.5$ | $-14.0 \pm 2.8$ | $-16.8 \pm 2.9$ | $-14.1 \pm 1.8$ | $-13.4 \pm 4.9$ |
| DemonAttack | $162.0 \pm 34.7$ | $386.2 \pm 75.3$ | $183.2 \pm 44.7$ | $372.9 \pm 81.7$ | $570.3 \pm 110.2$ |

Table 4: Mean and standard deviation of returns per Atari game across 5 random seeds using 1% of replay dataset after 6.25M gradient steps. REM and CQL results are from Kumar et al. (2022).

| Game | REM | CQL | AEVL | Fixed-CCVL | CCVL |
|---|---|---|---|---|---|
| Asterix | $2317.0 \pm 838.1$ | $3318.5 \pm 301.7$ | $1958.9 \pm 1050.2$ | $3256.6 \pm 395.1$ | $5517.2 \pm 1215.4$ |
| Breakout | $33.4 \pm 4.0$ | $166.0 \pm 23.1$ | $16.7 \pm 5.6$ | $150.3 \pm 17.8$ | $172.5 \pm 35.6$ |
| Pong | $-0.7 \pm 9.9$ | $17.9 \pm 1.1$ | $-0.2 \pm 4.7$ | $17.6 \pm 2.1$ | $17.4 \pm 2.8$ |
| Seaquest | $2753.6 \pm 1119.7$ | $2030.7 \pm 822.8$ | $2853.0 \pm 1089.2$ | $2112.5 \pm 856.4$ | $2746.0 \pm 1544.2$ |
| Qbert | $7417.0 \pm 2106.7$ | $9605.6 \pm 1593.5$ | $5409.2 \pm 3256.6$ | $9750.7 \pm 1366.8$ | $10108.1 \pm 2445.5$ |
| SpaceInvaders | $443.5 \pm 67.4$ | $1214.6 \pm 281.8$ | $450.2 \pm 101.3$ | $1243.4 \pm 269.8$ | $1154.6 \pm 302.1$ |
| Zaxxon | $1609.7 \pm 1814.1$ | $4250.1 \pm 626.2$ | $1678.2 \pm 1425.6$ | $4060.3 \pm 673.1$ | $6470.2 \pm 1512.2$ |
| YarsRevenge | $16930.4 \pm 2625.8$ | $17124.7 \pm 2125.6$ | $17233.5 \pm 2590.8$ | $18040.5 \pm 1545.9$ | $19233.0 \pm 1719.2$ |
| RoadRunner | $46601.6 \pm 2617.2$ | $38432.6 \pm 1539.7$ | $45035.2 \pm 3823.0$ | $37945.7 \pm 1338.9$ | $42780.5 \pm 4112.3$ |
| MsPacman | $2303.1 \pm 202.7$ | $2790.6 \pm 353.1$ | $2148.8 \pm 273.4$ | $2501.5 \pm 201.3$ | $2680.4 \pm 212.4$ |
| BeamRider | $674.8 \pm 21.4$ | $785.8 \pm 43.5$ | $662.9 \pm 50.7$ | $782.3 \pm 34.9$ | $780.1 \pm 40.8$ |
| Jamesbond | $130.5 \pm 45.7$ | $96.8 \pm 43.2$ | $152.2 \pm 58.2$ | $112.3 \pm 81.3$ | $172.1 \pm 153.9$ |
| Enduro | $1583.9 \pm 108.7$ | $938.5 \pm 63.9$ | $1602.7 \pm 135.5$ | $913.2 \pm 50.3$ | $1376.2 \pm 203.8$ |
| WizardOfWor | $2661.6 \pm 371.4$ | $612.0 \pm 343.3$ | $1767.5 \pm 462.1$ | $707.4 \pm 323.2$ | $2723.1 \pm 515.6$ |
| IceHockey | $-6.5 \pm 3.1$ | $-15.0 \pm 0.7$ | $-9.1 \pm 4.8$ | $-17.6 \pm 1.0$ | $-10.2 \pm 2.1$ |
| DoubleDunk | $-17.6 \pm 2.6$ | $-16.2 \pm 1.7$ | $-19.4 \pm 3.2$ | $-15.2 \pm 0.9$ | $-9.8 \pm 3.8$ |
| DemonAttack | $5602.3 \pm 1855.5$ | $8517.4 \pm 1065.9$ | $2455.3 \pm 1765.0$ | $8238.7 \pm 1091.2$ | $9730.0 \pm 1550.7$ |

Table 5: Mean and standard deviation of returns per Atari game across 5 random seeds using 5% of replay dataset after 12.5M gradient steps. REM and CQL results are from Kumar et al. (2022).

| Game | REM | CQL | AEVL | Fixed-CCVL | CCVL |
|---|---|---|---|---|---|
| Asterix | $5122.9 \pm 328.9$ | $3906.2 \pm 521.3$ | $7494.7 \pm 380.3$ | $3582.1 \pm 327.5$ | $7576.0 \pm 360.2$ |
| Breakout | $96.8 \pm 21.2$ | $70.8 \pm 5.5$ | $97.1 \pm 35.7$ | $75.8 \pm 6.1$ | $121.4 \pm 10.3$ |
| Pong | $7.6 \pm 11.1$ | $5.5 \pm 6.2$ | $7.1 \pm 12.9$ | $5.2 \pm 6.0$ | $13.4 \pm 6.1$ |
| Seaquest | $981.3 \pm 605.9$ | $1313.0 \pm 220.0$ | $877.2 \pm 750.1$ | $1232.6 \pm 379.3$ | $1211.4 \pm 437.2$ |
| Qbert | $4126.2 \pm 495.7$ | $5395.3 \pm 1003.64$ | $4713.6 \pm 617.0$ | $5105.5 \pm 986.4$ | $5590.9 \pm 2111.4$ |
| SpaceInvaders | $799.0 \pm 28.3$ | $938.1 \pm 80.3$ | $692.7 \pm 101.9$ | $860.5 \pm 77.3$ | $1233.4 \pm 103.1$ |
| Zaxxon | $0.0 \pm 0.0$ | $836.8 \pm 434.7$ | $902.5 \pm 895.2$ | $904.1 \pm 560.1$ | $1212.2 \pm 902.1$ |
| YarsRevenge | $11924.8 \pm 2413.8$ | $12413.9 \pm 2869.7$ | $12508.5 \pm 1540.2$ | $11587.2 \pm 2676.8$ | $12502.6 \pm 2349.2$ |
| RoadRunner | $49129.4 \pm 1887.9$ | $45336.9 \pm 1366.7$ | $50152.9 \pm 2208.9$ | $44832.6 \pm 1329.8$ | $47972.1 \pm 2991.3$ |
| MsPacman | $2268.8 \pm 455.0$ | $2427.5 \pm 191.3$ | $2515.5 \pm 548.0$ | $2115.3 \pm 108.9$ | $2015.7 \pm 352.8$ |
| BeamRider | $4154.9 \pm 357.2$ | $3468.0 \pm 238.0$ | $4564.7 \pm 578.4$ | $3312.3 \pm 247.3$ | $3781.0 \pm 401.8$ |
| Jamesbond | $149.3 \pm 304.5$ | $89.7 \pm 15.6$ | $127.6 \pm 414.8$ | $91.9 \pm 20.2$ | $152.8 \pm 42.8$ |
| Enduro | $832.5 \pm 65.5$ | $1160.2 \pm 81.5$ | $959.2 \pm 100.3$ | $1204.6 \pm 90.3$ | $1585.0 \pm 102.1$ |
| WizardOfWor | $920.0 \pm 497.0$ | $764.7 \pm 250.0$ | $1184.3 \pm 588.9$ | $749.3 \pm 231.8$ | $1429.9 \pm 751.4$ |
| IceHockey | $-5.9 \pm 5.1$ | $-16.0 \pm 1.3$ | $-5.2 \pm 7.3$ | $-14.9 \pm 2.5$ | $-4.1 \pm 5.9$ |
| DoubleDunk | $-19.5 \pm 2.5$ | $-20.6 \pm 1.0$ | $-19.2 \pm 2.2$ | $-21.3 \pm 1.7$ | $-24.6 \pm 6.2$ |
| DemonAttack | $9674.7 \pm 1600.6$ | $7152.9 \pm 723.2$ | $10345.3 \pm 1612.3$ | $7416.8 \pm 1598.7$ | $12330.5 \pm 1590.4$ |

Table 6: Mean and standard deviation of returns per Atari game across 5 random seeds using initial 10% of replay dataset after 12.5M gradient steps. REM and CQL results are from Kumar et al. (2022).

