# OpenReview forum: "Confidence-Conditioned Value Functions for Offline Reinforcement Learning"
_NeurIPS.cc/2022/Workshop/Offline_RL — Offline RL Workshop NeurIPS 2022_

### Official Review · Reviewer_CP5z · 2022-10-19

**Rating:** 6
**Confidence:** 4

**Review:**

This paper proposes CCVL, an offline RL algorithm for discrete-action environments that can adjust their level of conservatism during online rollouts. The method and mathematical derivation are sound, though I did not check the math in more detail.

The related work should discuss relation to CODAC (https://arxiv.org/abs/2107.06106), which learns a conservative distributional Q-function that can adjust to user specified risk-level. On this note, I think the adaptive nature of CCVL can be better showcased by considering setups where this adaptiveness is useful or necessary. The Atari experiments, though standard, are not motivated or tailored to this method's particular strength. The online fine-tuning experiment is more compelling, but the results seem only significant in one task (Asterix).

---

### Official Review · Reviewer_FGbC · 2022-10-19
**Interesting work, but may benefit from explaining the difference between confidence-conditioned (this work) and some older uncertainty based works**

**Rating:** 6
**Confidence:** 5

**Review:**

This work propose using confidence conditioned value functions to produce pessimistic value estimate

Questions:
0. Could you report numbers on the D4RL benchmark like Table 1 in CQL?
1. Since the work is on confidence, have you considered uncertainty-based works like as baseline [1,2]? Also, some prior works have been using quantile based networks for online learning.
2. Have you considered giving some empirical analysis on the confidence scores generated by the model?

[1] Wu, Yue, et al. "Uncertainty weighted actor-critic for offline reinforcement learning." arXiv preprint arXiv:2105.08140 (2021).
[2] Bai, Chenjia, et al. "Pessimistic bootstrapping for uncertainty-driven offline reinforcement learning." arXiv preprint arXiv:2202.11566 (2022).